# The Transmembrane Protease TMPRSS2 as a Therapeutic Target for COVID-19 Treatment

**DOI:** 10.3390/ijms23031351

**Published:** 2022-01-25

**Authors:** Lukas Wettstein, Frank Kirchhoff, Jan Münch

**Affiliations:** Institute of Molecular Virology, Ulm University Medical Center, 89081 Ulm, Germany; lukas.wettstein@uni-ulm.de (L.W.); frank.kirchhoff@uni-ulm.de (F.K.)

**Keywords:** TMPRSS2, coronavirus, SARS-CoV-2, influenza, protease inhibitor

## Abstract

TMPRSS2 is a type II transmembrane protease with broad expression in epithelial cells of the respiratory and gastrointestinal tract, the prostate, and other organs. Although the physiological role of TMPRSS2 remains largely elusive, several endogenous substrates have been identified. TMPRSS2 serves as a major cofactor in SARS-CoV-2 entry, and primes glycoproteins of other respiratory viruses as well. Consequently, inhibiting TMPRSS2 activity is a promising strategy to block viral infection. In this review, we provide an overview of the role of TMPRSS2 in the entry processes of different respiratory viruses. We then review the different classes of TMPRSS2 inhibitors and their clinical development, with a focus on COVID-19 treatment.

## 1. Type II Transmembrane Serine Proteases

Proteases initiate and regulate numerous fundamental physiological processes through precise and timely processing of proteins and peptides [1,2]. About one third of proteases belong to the family of serine proteases [3]. While the archetypical serine proteases, such as trypsin or thrombin are secreted from cells, many proteases are membrane-associated [4]. Type II transmembrane serine proteases (TTSPs) are multidomain enzymes characterized by an N-terminal intracellular domain, a transmembrane domain, an extracellular variable stem region, and a C-terminal serine protease domain. Today, the TTSP family includes more than twenty members, which are divided into four subfamilies depending on the phylogeny and architecture of their stem region: Hepsin/TMPRSS, Matriptase, HAT/DESC, and Corin [5]. Herein, we discuss the physiological role of the transmembrane protease serine subtype 2 (TMPRSS2) and its utilization as an entry cofactor in viral infection and a target for antiviral agents (Figure 1).

## 2. Basic Features of TMPRSS2

### 2.1. Identification and Structure

In 1997, Paoloni-Giacobino and colleagues described a novel 492 amino acid (aa) transmembrane protease named TMPRSS2, belonging to the Hepsin/TMPRSS subfamily of TTSPs [6]. TMPRSS2 is characterized by an N-terminal intracellular domain, a type II transmembrane domain, a stem region comprising LDLRA (LDL receptor class A) and SRCR (scavenger receptor cysteine-rich) domains, and a C-terminal serine protease domain (Figure 2) [6].

The contribution of the single domains to TMPRSS2 function has not been fully elucidated. Experimental analysis of TMRPSS2 mutants suggests the involvement of the LDLRA domain in enzymatic activity [7]. Consistent with this finding, the LDLRA domain contributes to the activation of the related TTSPs matriptase and matriptase-2 [8,9]. To date, the exact role of the SRCR domain in TMPRSS2 is unclear. However, in general SRCR domains appear to be involved in interactions with the cell surface or extracellular molecules [6,10,11]. Analysis of TTSP TMPRSS3 mutants suggests that the SRCR domain has a role in proteolytic activity [12]. The serine protease domain comprises a catalytic triad consisting of histidine (H296), aspartate (D345) and serine (S441). It preferentially cleaves substrates with a monobasic arginine (R) or lysine (K) residue at the P1 position [6,13,14]. Similar to other TTSPs, TMPRSS2 is expressed as single chain zymogen that undergoes autoproteolytic cleavage at the interface of the SRCR– and serine protease domains (R255↓I256) to acquire proteolytic activity [15]. The protease domain either remains linked to the prodomain via an interdomain disulfide bond or is shed, resulting in the membrane-bound and cell-free forms, respectively [15,16,17].

### 2.2. Expression

The human *TMPRSS2* gene is located on chromosome 21 (21q22.3) and comprises 14 exons and 13 introns [6,18]. Genetic analysis revealed the presence of an androgen response element (ARE) in the 5′ untranslated region (UTR) of the *TMPRSS2* gene [18,19]. Consequently, its transcription can be regulated by androgenic hormones [17,18,19]. Alternative splicing of TMPRSS2 mRNA may give rise to a second isoform that differs from the 492 aa variant by an elongated (37 aa) cytoplasmic tail [18,20]. There is no evidence for any differential proteolytic activities or physiological roles of the isoforms. For this reason, the vast majority of studies have focused on the 492 aa isoform. TMPRSS2 is expressed in a variety of human organs and tissues including the prostate, respiratory tract, and gastrointestinal tract, as well as the liver, kidney, pancreas, thymus and salivary glands [6,15,18,19,21,22]. In healthy prostate, TMPRSS2 is predominantly localized on the apical membrane of luminar epithelial cells [17,21,23]. However, prostate cancer is associated with overexpression and a more diffuse localization pattern of TMPRSS2 [19,23,24]. In the respiratory tract, transcripts of TMPRSS2 have been detected in the epithelial cells of nasal and tracheal tissues and in the distal airways [25]. Immunostaining of tissue samples confirms the broad expression of TMPRSS2 in upper airway epithelia, as well as in bronchial and alveolar epithelium [22]. Of note, TMPRSS2 protein expression in alveoli has been detected in type II (ATII) but not type I (ATI) alveolar cells [22]. Single cell RNA sequencing analyses indicate strong expression of TMPRSS2 across lung cell types, preferentially in ATII but also in ATI cells [26,27]. Notably, results on the androgen dependency of TMPRSS2 expression in the lung are contradictory: upregulation of TMPRSS2 has been reported in human lung cells after testosterone treatment; however, TMPRSS2 was not listed among the genes upregulated by testosterone in murine lungs [28]. Similarly, treatment of mice with the androgen inhibitor enzalutamide did not result in reduced TMPRSS2 transcription, suggesting androgen-independent expression in mouse lungs [29]. Whether this observation applies to human tissue remains to be elucidated.

### 2.3. Endogenous Substrates

The physiological role of TMPRSS2 remains largely elusive. Kim and colleagues described the generation of *Tmprss2^-/-^* deficient mice for the serine protease domain. These mice exhibited neither phenotypic abnormalities in embryonic development, fertility or organ function nor elevated expression levels of TTSP homologs [16]. The authors speculated that functional redundancy with homologous TTSP or the presence of the TMPRSS2 stem domain compensates for the lack of protease activity, and suggested analysis of mice with multiple knockouts or additional disruption of the TMPRSS2 stem region, respectively [16]. Several potential substrates for TMPRSS2 have been described. Upon expression of TMPRSS2, Donaldson and colleagues observed a reduced sodium ion current mediated by the epithelial sodium channel (ENaC) [25]. Although this suggests protease-dependent regulation of the ENaC, it remains unclear whether this is a TMPRSS2-specific phenomenon. In vitro studies using an androgen-dependent prostate cancer cell line identified protease-activated receptor-2 (PAR-2) as a potential endogenous substrate for TMPRSS2 [30]. Addition of the protease resulted in activation of PAR-2 and release of calcium ions from intracellular stores, indicating initiation of downstream signaling [30,31]. Subsequent studies revealed that overexpression of TMPRSS2 activates the TTSP matriptase in prostate cancer cells, which in turn activates PAR-2 [32,33]. Although PAR-2 is expressed in a variety of TMPRSS2-positive tissues and mediates protective functions in the gastrointestinal and respiratory tract, the role of TMPRSS2 in the matriptase-PAR-2 axis requires further investigation [34,35,36,37]. Considering the TMPRSS2 substrate preference, an in silico screen identified serine protease-like zymogens as potentially activated by TMPRSS2 and verified the TMPRSS2-dependent cleavage of pro-kallikrein-2 (KLK2, Uniprot P20151) to mature KLK2 [14]. KLK-2 is secreted by the prostate and contributes to the anticoagulation of seminal fluid by activating prostate-specific antigen [38]. Furthermore, TMPRSS2 has been shown to activate the single-chain precursor of hepatocyte growth factor (HGF), which contributes to prostate cancer metastasis [14]. In general, several lines of evidence suggest an involvement of TMPRSS2 activity in the progression of prostate cancer [14,23,24,32]. The relatively high expression levels in prostate tissue and the identification of TMPRSS2 substrates in prostate cancer models argue for a potential role of TMPRSS2 in regular prostate function. However, the relevance and effects of TMPRSS2 activity on the physiological function of the prostate in vivo remain to be determined. Studies of TMPRSS2 in systems not associated with prostate malignancies, such as gastrointestinal and respiratory models, might contribute to unravelling the physiological impact of TMPRSS2—a challenging task given the promiscuity and functional redundancy of many human proteases. In contrast to its physiological role, the involvement of TMPRSS2 in viral infection has been extensively investigated.

## 3. Role of TMPRSS2 in Viral Entry

Enveloped viruses such as corona or influenza viruses enter host cells via the binding of their envelope glycoproteins (GP) to their cognate receptors. To trigger fusion between the viral and cellular membranes, some viral GPs rely on proteolytic activation. The timing and location of proteolytic activation varies between different viruses. While some GPs become activated during viral egress from a producer cell, others are cleaved immediately before fusion with a new host cell, or depend on two independent cleavage events [7,39]. The preference for certain proteases is governed by the presence of specific cleavage motifs in the GPs, and changes in these motifs can significantly alter viral tropism and infectivity [7]. As described below, this phenomenon has been described for several respiratory viruses. TMPRSS2 is widely expressed in the human airways, and its role in the activation of clinically relevant respiratory viruses such as influenza and coronaviruses is well established [40].

### 3.1. Influenza Virus

Influenza viruses A and B carry two GPs, neuraminidase and hemagglutinin (HA). HA is synthesized as a single chain precursor protein (HA0) and encompasses two subunits, referred to as HA1 and HA2 (Figure 3a). In order to acquire fusion competence, HA depends on proteolytic cleavage of a linker at the HA1–HA2 interface, a process that triggers conformational rearrangements and enables pH dependent exhibition of the viral fusion peptide [41]. TMPRSS2-mediated cleavage has been described mainly for influenza A viruses; in accordance with the substrate preference of the proteases, cleavage occurs preferentially after a monobasic motif in the linker region [42]. TMPRSS2 was identified as an HA-activating protease in 2006 when Böttcher and colleagues demonstrated TMPRSS2- and HAT- (human airway trypsin-like protease, TMPRSS11D)dependent activation of human-pathogenic influenza strains H1N1, H2N9 and H3N2 [43]. Further in vitro studies showed TMPRSS2-dependent cleavage of various HA subtypes with monobasic cleavage sites (H1, H2, H4, H6, H8–11, H13, H14 and H16), although this process is not exclusively TMPRSS2-dependent [43,44,45,46]. Proteases such as the TTPs TMPRSS4, hepsin, matriptase and HAT have been described as activating proteases for certain HA proteins [44,45,46,47,48]. Experiments on canine cells revealed a preferential activation of HA by HAT at the cell surface, whereas TMPRSS2-dependent activation occurred in the secretory pathway during viral egress [49].

Infection of TMPRSS2 knockout mice resulted in reduced pathogenicity and replication of monobasic H1N1 or H7N9 and H10-bearing influenza virus strains, as compared to infection of WT mice, thus confirming the TMPRSS2 dependence of these strains [50,51,52,53,54]. Interestingly, TMPRSS2 sensitivity of the monobasic strain H3N2 varied greatly throughout these in vivo studies. This observation might be due to differences in infection protocols, acquisition of an escape mutation in H3N2 HA that allows TMPRSS2 independent cleavage, or activation of H3 HA by the TTSP TMPRSS4 [48,50,51,53,55]. Limburg and colleagues observed TMPRSS2-independent spread of H3N2 in alveolar cells of TMPRSS2-deficient mice, whereas knockdown of TMPRSS2 in primary human alveolar cells prevented replication of this strain [56]. They hypothesized that differences in the protease repertoires of the murine and human genome account for this observation [56]. Similar to H3, the importance of TMPRSS2-mediated activation of influenza B virus (IBV) HA varies depending on the cell culture or animal model used in the study. On the one hand, knockdown of TMPRSS2 did allow replication of IBV in Calu-3 or primary human bronchiolar epithelial cells, as well as in TMPRSS2 knock-out mice [52,56]; however, on the other hand IBV failed to replicate in TMPRSSS2-negative MDCK or in primary type II alveolar epithelial cells with TMPRSS2 knockdown [52,56,57].

In contrast to strains with a monobasic cleavage site, highly pathogenic influenza A viruses often harbor a multibasic cleavage site and depend on cleavage by proprotein convertases (PPCs) such as furin [41]. Consequently, replication of these multibasic strains occurs independently of TMPRSS2 both in vitro and in vivo [46,50,53]. PPCs are expressed ubiquitously, thus contributing to the systemic spread and pathogenicity of the virus [41,58]. Interestingly, TMPRSS2 was shown to activate mono- and multibasic HAs of H9N2 influenza strains, which are not activated by furin [47]. Overall, TMPRSS2 is only a suitable drug target for influenza virus strains with a monobasic, not a polybasic HA cleavage site.

### 3.2. Human Coronaviruses

Human coronaviruses (hCoVs) are a family of enveloped respiratory viruses. Of the seven known hCoVs, the common cold CoVs 229E, OC43, NL63 and HKU1 usually cause mild respiratory illness, while the SARS- (severe acute respiratory syndrome) and MERS- (middle eastern respiratory syndrome) CoV, as well as the currently pandemic SARS-CoV-2 can cause severe disease [39]. The coronavirus spike (S) GP is synthesized as an inactive precursor consisting of two subunits: S1, which contains the receptor binding domain, and S2, which harbors the viral fusion machinery (Figure 3b) [39]. In contrast to influenza virus HA, hCoV S proteins harbor two distinct protease cleavage sites, one located at the interface of the subunits, termed S1/S2, and another within the S2 subunit, termed S2’. Coronavirus entry is proposed to occur in a sequential manner: initial cleavage at the S1/S2 site is thought to prime the S protein for receptor binding and exposure of the S2’ site. Subsequently, receptor binding can be initiated, followed by cleavage of the S2’ site, which is crucial for triggering the fusion event, as shown for SARS-CoV-2 (Figure 3c) [59,60,61].

#### 3.2.1. SARS-CoV

The SARS-CoV S protein contains two monobasic cleavage sites. SARS-CoV infection requires the expression of the ACE2 receptor and depends on proteolytic activation mediated by cathepsins in many cell types, arguing for an entry via the endosomal route [62,63]. However, experiments with cells expressing TMPRSS2 allowed SARS-CoV entry in the presence of cathepsin inhibitors, suggesting that TMPRSS2 acts as a SARS-CoV S activating protease [64,65,66] for cathepsin-independent entry (Figure 3b). Both cleavage sites of SARS-CoV are most likely primed at the cell surface prior to fusion [65,67].

#### 3.2.2. MERS-CoV

Priming of MERS-CoV glycoproteins occurs during viral egress from the producer cell and is mediated by the proprotein convertase furin, which recognizes a multibasic cleavage site at the S1/S2 interface [60,68,69]. Additionally, MERS-CoV S is activated by the action of endosomal cathepsins and TMPRSS2 at the S2´ site [68,70,71,72]. Although the MERS-CoV spike S2´ site contains a multibasic minimal furin recognition sequence, cleavage of the S2´ site is independent of furin [68,73]. A precleaved S1/S2 subunit facilitates activation of the S2´ site by TMPRSS2, although S1/S2 cleavage is not mandatory for TMPRSS2-mediated entry [68,69,72].

#### 3.2.3. SARS-CoV-2

Due to its high sequence similarity to SARS-CoV, SARS-CoV-2 was rapidly found to infect cells expressing ACE2, whereby proteolytic activation of the S protein can occur via TMPRSS2 or cathepsins B and L [74]. In contrast to SARS-CoV, the S1/S2 site of SARS-CoV-2 S harbors a multibasic cleavage site with a minimum furin recognition motif [75]. Indeed, cleavage of S1/S2 site has been shown to be mediated by furin during viral egress [13,76]. This process is required for TMPRSS2 activation of S at the S2´ site upon receptor binding and is crucial for infecting cells of the respiratory tract [13,76,77,78,79]. Of note, in addition to TMPRSS2 several other TTSPs were able to prime SARS-CoV-2 S in vitro [80,81,82]. Although based on their expression levels TMPRSS11D, TMPRSS11E and TMPRSS11F as well as TMPRSS13 are unlikely to act in the lung, they might contribute to the extrapulmonary spread of SARS-CoV-2 [83]. It would be intriguing to gain more insight into the role of these proteases on SARS-CoV-2 infection.

#### 3.2.4. Common Cold Coronaviruses

For common cold CoVs, proteolytic activation is best studied for 229E. Entry of 229E can occur via cathepsin- or TMPRSS2-dependent pathways [84,85,86]. However, it has been shown that clinical isolates of common cold CoVs 229E, as well as OC43 and HKU1, are more susceptible to inhibition of TMPRSS2 than to inhibition of cathepsins, suggesting that these viruses preferentially employ a TMPRSS2-dependent route of entry [85,87,88]. In the case of NL63, the impact of pH and cathepsin dependence during entry remains under debate [89,90]. However, NL63 displays increased entry in TMPRSS2-expressing cells and is susceptible to TMPRSS2 inhibition [66,89,90,91,92]. In sum, this suggests an influence of TMPRSS2 on the entry of common cold CoVs.

Altogether, coronavirus fusion can occur at the cell membrane, as well as in endosomal compartments, and may be triggered by cathepsins or by membrane protease, such as TMPRSS2. For comprehensive reviews of this topic, see references [93,94]. Although both modes of entry can occur in cell culture, several lines of evidence suggest a preference towards TMPRSS2-dependent fusion during in vivo infection. Coronaviruses primarily replicate in the respiratory tract, although expression levels of cathepsins in the lungs were insufficient to support priming of MERS-CoV [72]. As mentioned above, clinical isolates of common cold coronaviruses preferentially enter cells via TMPRSS2, and cathepsins appear dispensable for entry of SARS-CoV-2 into lung cells [85,87,88]. Moreover, the pathogenicity and spread of SARS-CoV, MERS-CoV and SARS-CoV-2 was reduced in TMPRSS2 knockout mice, although viral replication was not completely suppressed [29,95]. In rodents, administration of the serine protease inhibitors camostat and nafamostat reduced the pathogenicity and replication of SARS-CoV and SARS-CoV-2 [96,97]. Hence, targeting the activity of TMPRSS2 with inhibitors might be a promising approach to control coronavirus infection.

### 3.3. Parainfluenza, Metapneumovirus and Sendaivirus

TMPRSS2 not only activates the GPs of coronaviruses and influenza viruses, it also activates the fusion (F) GP of human parainfluenza viruses (HPIV) 1, 2, 3, 4a and 4b and human metapneumovirus (HMPV) in cell culture [98,99]. Both viruses are major causes of acute lower respiratory illness (ALRI) in young children [100]. Furthermore, the F protein of sendaivirus (SeV) can be cleaved by TMPRSS2, and presence of the protease accelerates spread of the virus [98].

## 4. Inhibition of TMPRSS2 Function

The involvement of TMPRSS2 in respiratory viral infection makes this protease an attractive therapeutic target [40,101,102]. Pharmacological inhibition of TMPRSS2 is supported by the absence of phenotypic abnormalities upon TMPRSS2 knockout in mice [16]. In the last decade, a broad spectrum of TMPRSS2 inhibitors has been described, encompassing small molecule compounds, peptides/proteins, and peptidomimetics (Figure 4). The emergence and pandemic spread of the novel coronavirus SARS-CoV-2 has further accelerated the search for inhibitors of TMPRSS2. Most TMPRSS2 inhibitors target its proteolytic activity, although agents aiming to reduce its expression or biosynthesis are being developed as well (Figure 4).

### 4.1. Inhibitors of TMPRSS2 Expression

Androgen deprivation therapy (ATD) is employed in prostate cancer treatment to reduce androgen-induced tumor growth. Consequently, reduction of androgen signaling reduces ARE-dependent transcription of TMPRSS2 in the prostate [103]. Commonly employed drug-based methods rely on chemical castration or treatment of patients with antiandrogens and other hormone agonists [104]. While this therapy exhibits therapeutic benefits in prostate cancer progression, it is associated with severe side effects and can be bypassed by the occurrence of androgen-independent prostate cancer [105,106]. As discussed above, the androgen dependence of TMPRSS2 expression in the human respiratory tract has not been fully elucidated [28,29]. Thus, ATD does not appear to be a suitable option for prevention and treatment of infection with TMPRSS2-dependent viruses.

In addition to broad-acting antiandrogens, several agents have been described that reduce TMPRSS2 expression in cell culture. Peptide-conjugated phosphorodiamidate morpholino oligomers (PPMOs) are cell-penetrating antisense DNA analogues that allow downregulation of protein biosynthesis at the level of mRNA maturation [107]. T-ex5, a PPMO interfering with the correct splicing of exon 5 in TMPRSS2 pre-mRNA, induces the expression of a TMPRSS2 protein deficient for the LDLRA domain [7]. This form is incapable of autoproteolytic activation and lacks proteolytic activity. Consequently, treatment with T-ex5 hampered the replication of monobasic influenza virus strains and SARS-CoV-2 S activation in cell lines and primary cells [7,13,42,56]. In an in vitro screening approach, Chen and colleagues identified therapeutic compounds that reduce TMPRSS2 protein expression in the low- to sub-micromolar range. The two most active compounds, halofuginone and homoharringtonine, significantly reduced SARS-CoV-2 replication in immortalized Calu-3 lung cells [108]. Mechanistically, halofuginone impairs TMPRSS2 protein stability by increasing proteasomal degradation and affects translation due to its function as glutamyl-prolyl-tRNA inhibitor [108,109]. Homoharringtonine interferes with protein biosynthesis through blockade of the large ribosomal subunit and has been shown to inhibit a variety of TMPRSS2-dependent (PEDV, Dong 2018) and -independent (VSV, HSV-1, RV; VZV, MHV) viruses [110,111,112]. However, whether the anti-SARS-CoV-2 activity is attributed to reduction of TMPRSS2 expression or due to generally reduced translation remains to be determined.

### 4.2. Inhibitors of TMPRSS2 Activity

#### 4.2.1. Small Molecule Compounds

Camostat mesylate (CM) is an inhibitor of trypsin-like serine proteases. Initially, CM was shown to inhibit the proteolytic activity of trypsin, plasmin, kallikrein, thrombin and C1-esterase [113]. Currently, it is approved for the treatment of chronic pancreatitis and postoperative reflux esophagitis in Japan [83,114,115]. An activity against TMPRSS2 was first discovered in the search for inhibitors of SARS-CoV S activation [92]. Subsequently, CM has been reported to reduce the infectivity of various respiratory viruses including SARS-CoV, MERS-CoV, NL63, 229E, OC43, HKU-1, and influenza virus at micromolar concentrations [66,70,71,74,85,87,88,91,92,97,116,117,118]. In line with these findings, CM efficiently inhibits TMPRSS2-dependent entry of SARS-CoV-2 [74,83,97,117,119,120,121,122,123,124]. Further in vitro studies on cell-associated or purified TMPRSS2 protein confirmed a reduction of TMPRSS2 protease activity by CM (Table 1) [83,120,123,124,125,126,127]. Moreover, oral CM treatment protected mice from SARS-CoV-induced mortality [96]. Upon oral or intravenous administration, esterases rapidly convert CM into its metabolite GBPA (4-(4- guanidinobenzoyloxy)phenylacetic acid) [128,129]. Although GBPA-mediated inhibition of TMPRSS2 protease activity is reduced up to ~10-fold compared to CM, there is no discernible difference in the anti-SARS-CoV-2 activity of the two compounds [83,127]. This is likely due to the efficient conversion of CM to GBPA under standard cell culture conditions [83].

By screening a drug library, Yamamoto et al. identified the protease inhibitor Nafamostat mesylate (NM) as an inhibitor of MERS-CoV fusion and reasoned that NM acts as inhibitor of TMPRSS2 (Table 1) [116]. Similar to CM, NM is a synthetic guanidinobenzoate derivative that inhibits various serine proteases, including TMPRSS2 [125,126,127,130]. NM is approved as an anticoagulant and for treatment of pancreatitis in Japan and Korea [97,131,132,133]. In direct comparison, NM inhibited recombinant TMPRSS2 ~10- times more potently than CM [125,126,127]. Consequently, NM inhibited the entry of SARS-CoV and SARS-CoV-2 pseudoparticles, as well as genuine MERS-CoV and SARS-CoV-2 infection, more efficiently than CM in primary and immortalized human cells [71,97,117,119,121,125,134,135]. Of note, SARS-CoV-2 variants of concern (VOC) Alpha, Beta or Gamma did not differ in their susceptibility towards either CM or NM compared to the early Wuhan SARS-CoV-2 strain [119,122,125]. Experiments on CM against S-dependent entry of SARS-CoV-2 VOC Delta confirmed this observation, while there is no data available for NM [122]. The potency of NM in curbing SARS-CoV-2 infection in vivo has been investigated using transgenic ACE2 mice or mice sensitized by transduction of ACE2. In both models, a single intranasal administration of NM at 3 mg/kg reduced viral loads in the lung, pathogenicity, and weight loss, particularly when treatment preceded SARS-CoV-2 infection [97].

The ester linkage of guanidinobenzoyl-derived inhibitors, such as CM or NM, is recognized as a scissile bond for serine proteases. However, the cleavage is incomplete and blocks the catalytic triad by forming a long-lived covalent acyl enzyme intermediate [136,137]. Because the crystal structure of TMPRSS2 has been solved only recently, molecular dynamics simulations based on homology models were performed to elucidate the binding mode of CM, NM and GBPA to the protease [138]. All three compounds were predicted to form non-covalent Michaelis complexes (MC) that accommodate the active site of TMPRSS2 in a manner favorable for initiating substrate cleavage. The guanidino moiety of the inhibitors interacts with the D435 of the S1 pocket, while the oxyanion hole of the proteases confers additional stability to the MC. Furthermore, the ester bond of the compounds is in close proximity to the serine and histidine residues of the catalytic triad [83,125,126]. In the case of NM, an inverted binding to the TMPRSS2 active site is possible as well [83,126]. Interestingly, the stability of the MC with TMPRSS2 is highest for NM, followed by CM and GBPA, reflecting the differences in their antiviral activity [126].

Similar to CM and NM, the small molecule protease inhibitor Gabexate was predicted to interact with the substrate binding site of TMPRSS2, although yielding a less stable complex [125]. There are conflicting results on the antiviral activity of Gabexate (Table 1). Even though it reduced the enzymatic activity of TMPRSS2, Hu and colleagues did not observe any reduction in SARS-CoV-2 S-mediated entry into lung cells [125,127]. Consistent with this, Gabexate did not affect cell–cell fusion mediated by MERS-CoV or SARS-CoV-2 or S-mediated entry of SARS-CoV or NL63 [92,116,134]. Similarly, Hoffmann and colleagues reported little if any reduction of SARS-CoV, SARS-CoV-2 and MERS-CoV S-mediated entry in lung cells [117]. Yamaya and colleagues found a reduction in influenza virus replication in primary human tracheal cells, while Kosai et al. did not observe reduced influenza viral loads in a mouse model [118,139]. Further studies under uniform experimental conditions would be required to clarify the impact of Gabexate on protease activity and viral replication. However, Gabexate clearly seems inferior to CM or NM in terms of anti-TMPRSS2 and antiviral activity.

Bromhexine hydrochloride (BHH) is an FDA-approved mucolytic cough suppressant [140]. In an in vitro screen for drugs targeting TMPRSS2, BHH was found to decrease the proteolytic activity of purified TMPRSS2 [14]. In contrast, another study using an analogous experimental approach reported no effect of BHH (Table 1) [127]. While both studies implied the same fluorogenic protease substrate as readout, the preparations of recombinant TMPRSS2 differed: the former study purified the TMPRSS2 extracellular domain devoid of the LDLRA domain (aa 148–492), whereas the latter study used TMPRSS2 and included the entire extracellular domain (aa 106–492) [14,127]. It is unclear whether this explains the observed discrepancies in BHH-mediated inhibition of TMPRSS2; experiments with full length cellular TMPRSS2 may clarify this. Consistent with the role of TMPRSS2 in prostate cancer, Lucas and colleagues reported a reduction in prostate cancer metastasis upon treatment with BHH in vitro and in vivo [14]. BHH completely inhibited the infection of authentic SARS-CoV-2 in TMPRSS2-negative cells, and was only slightly active in TMPRSS2-expressing lung cells [119]. On the other hand, CM and NM displayed full antiviral activity only in lung cells [119]. Thus, this argues that BHH exerts its anti-SARS-CoV-2 activity by inhibition of cathepsin rather than TMPRSS2 The same screening that identified BHH revealed four additional substances (labelled 0591-5323, 4401-0077, 4554-5138, 8008-1235) that reduced the activity of TMPRSS2 in the low micromolar range [14]. However, due to low availability of the compounds these hits were not used for follow-up analysis.

Other small molecule compounds were identified in two in silico screenings for TMPRSS2 inhibitors: the factor Xa inhibitor Otamixaban, the urokinase-type plasminogen activator UKI-1, the kallikrein inhibitor Avoralstat, and the factor VIIa inhibitor PCI-27483 all decreased TMPRSS2 activity (Table 1) [120,125]. However, only PCI-27483 and Avoralstat were shown to efficiently curb SARS-CoV-2-mediated entry and infection, with avoralstat being the more potent inhibitor [120]. Subsequent infection studies on hACE2-transduced BALB/c mice revealed that intraperitoneal administration of Avoralstat reduces SARS-CoV-2 titer in the lung and weight loss in a similar manner to CM [120].

#### 4.2.2. Peptides and Proteins

Aprotinin is a single chain 58 aa polypeptide from bovine lung that inhibits several serine proteases, including trypsin, chymotrypsin and plasmin [141,142]. Experiments with TMPRSS2-expressing MDCK cells revealed a reduction of influenza virus infection with aprotinin [143]. Moreover, aprotinin suppressed infection and replication of influenza virus in human cell lines, chicken embryonated eggs, primary human tracheal and adenoid cells, and mice by impeding HA0 activation [118,144,145,146,147,148]. However, aprotinin targets various proteases, many of which, such as HAT or TMPRSS4, are involved in the activation of influenza virus [44,45,46,47,48,143]. Aprotinin was shown to ameliorate the SARS-CoV-2 cytopathic effect and replication in immortalized cells and to reduce viral gene expression in primary bronchial epithelial cells [119,149]. SARS-CoV-2 replication kinetics in lung cells revealed an inhibitory effect of aprotinin on infectious virus yield over a 48 h period, while the antiviral effect vanished at 72 h post-infection [13]. Of note, aerosolized aprotinin reduced influenza virus loads in infected mice, and aerosolized administration has been approved in Russia [144,150]. Taken together, the antiviral properties of aprotinin confirm its inhibitory effect on TMPRSS2, even though direct evidence using recombinant protease is lacking (Table 1). Based on computational modelling, the oligopeptide antipain and the protein soy bean trypsin inhibitor (SBTI) were predicted to interact with TMPRSS2 [120]. In line with this prediction, both antipain and SBTI inhibited the activity of purified TMPRSS2, as well as the entry and genome replication of SARS-CoV-2 [120]. However, SBTI failed to inhibit the spread of influenza virus in canine cells expressing TMPRSS2 [143].

In search of endogenous inhibitors of TMPRSS2, Ko and coworkers performed co-immunoprecipitation assays on prostate cancer cells and identified HGF activator inhibitor (HAI)-1 and HAI-2 as interaction partners of TMPRSS2 (Table 1) [135]. HAI-1 and HAI-2 are transmembrane Kunitz-type protease inhibitors that are expressed in various tissues, including the prostate and the respiratory tract [151,152]. Both HAI-1 and HAI-2 reduced TMPRSS2 protease activity in vitro, with HAI-2 being ~100-fold more potent than HAI-1 [135]. Consequently, HAI-2 overexpression reduced TMPRSS2-mediated cleavage of HGF, c-Met signaling, and thus prostate tumor growth and metastasis [135]. HAI-2 reduced the growth of influenza virus and infection of human parainfluenzavirus-1 in both TMPRSS2-expressing and TMPRSS2-negative cells, indicating that additional proteases involved in viral entry may be inhibited [153,154].

α_1_ antitrypsin (α_1_AT) belongs to the serine protease inhibitor (serpin) superfamily, is primarily involved in maintaining the protease–antiprotease balance in the lung, and possesses anti-inflammatory properties [155,156]. Recently, α_1_AT was identified as an endogenous inhibitor of SARS-CoV-2 infection in human colon and lung cells [123,124]. A third study reported no influence of α_1_AT on SARS-CoV-2 infection [149], which might be attributed to the use of a ten times higher viral inoculum together with a ten-fold lower concentration of α_1_AT. Further analyses demonstrated an interaction of serpin and TMPRSS2, and showed that α_1_AT inhibits the protease activity of recombinant and cellular TMPRSS2 in the micro- to nanomolar range (Table 1) [123,124]. α_1_AT levels are upregulated up to five-fold during the acute phase response during inflammation [155]. Interestingly, an increased IL-6:α_1_AT ratio has been associated with a worsening of the clinical course of COVID-19 [157]. The incidence of COVID-19-related deaths has also been correlated with the occurrence of α_1_AT-deficiency, a genetic disorder that results in reduced serum α_1_AT levels [158]. Although confounding factors may affect this correlation, the antiviral, immunomodulatory and anti-inflammatory properties of α_1_AT render this serpin a promising drug for treatment of COVID-19 [159,160,161]. Of note, plasma-purified α_1_AT is an approved drug for the treatment of α_1_AT deficiency, and has been shown to be safe when administered at high doses intravenously or by inhalation [162,163,164,165,166]. Several clinical studies on the use of α_1_AT for the treatment of COVID-19 have been initiated (Table 2).

#### 4.2.3. Peptidomimetics

Peptidomimetics are compounds whose pharmacophore mimics a natural peptide and retains the ability to interact with the biological target. Based on previously designed peptidomimetic inhibitors of the TTSPs HAT, matriptase, and matriptase 2, as well as other serine-like proteases, Meyer and colleagues described the development of a series of TMPRSS2 substrate analogues with inhibition constants (K_i_) in the low nanomolar range [167,168,169,170,171,172,173,174]. Compound MI-001 (referred to as BAPA, Benzoylsulfonyl-d-arginine-proline-4-amidinobenzylamide) was the most active derivate of a group of 4-aminobenzylamide inhibitors, with a K_i_ of 19–20 nM against recombinant TMPRSS2 [57,174]. A second group of inhibitors was based on 3-amidinophenylalanyl-derviatives. Of these, compound MI-432 (referred to as compound 92) inhibited recombinant TMPRSS2 with a K_i_ of 0.9 nM, while the structurally related compound MI-1900 (referred to as compound 113) displayed a K_i_ of 3 nM [174]. MI-001 and MI-432 reduced cleavage of influenza virus HA, thus suppressing the replication of influenza A strains in human lung cells and organ cultures of mouse trachea [57,175]. MI-001 reduced replication of influenza virus H1N1 and IBV, although it had no impact on the replication of multibasic influenza strain H7N1 (Table 1) [57]. More recently, Bestle and colleagues showed a 35- to 280-fold reduction of SARS-CoV-2 infectious virus yields when treating Calu-3 cells with either MI-432 or MI-1900 (Table 1) [13]. Similar to the small molecule inhibitors and proteins discussed, these peptidomimetic compounds have an affinity for serine proteases other than TMPRSS2. The presence of a proline residue at the P2 position in MI-001, for example, increases its anti-TMPRSS2 activity, but confers lower selectivity toward proteases such as factor Xa, thrombin, and related TTSPs [167,174,176]. Therefore, the physiological tolerability of the respective inhibitor must be closely monitored and considered during clinical development.

**Table 1 ijms-23-01351-t001:** Anti-TMPRSS2 and antiviral activity of inhibitors. ✓: Experimentally confirmed anti-TMPRSS2 or antiviral activity; 🗴: Experimentally refuted anti-TMPRSS2 or antiviral activity; ~: Experimentally confirmed marginal anti-TMPRSS2 or antiviral activity; n.a.: no data available.

			Antiviral Activity			
	Compound	Anti-TMPRSS2	Influenza	SARS-CoV	MERS-CoV	SARS-CoV-2
Small molecule	Camostat	✓ [120,123,124,125,126,127]	✓ [118]	✓ [74,96,117]	✓ [[76],[77],[80],[100], [113][114]]	✓ [74,83,97,117,119,120,121,122,123,124,125]
	GBPA (FOY251)	✓ [83,127]	n.a.	n.a.	n.a.	✓ [83]
	Nafamostat	✓ [125,126,127]	n.a.	✓ [117]	✓ [97,116,117]	✓ [86,100,113,115,116,118,119]
	Gabexate	✓ [125,127]	✓/🗴 [118,139]	✓/🗴 [92,117]	✓/🗴 [116,117,125]	~/🗴 [117,125,134]
	Bromhexine	✓/🗴 [14,125]	n.a.	n.a.	n.a.	✓ [119]
	0591-5323, 4401-0077, 4554-5138, 8008-1235	✓ [14]	n.a.	n.a.	n.a.	n.a.
	Otamixaban	✓ [125]	n.a.	n.a.	n.a.	~ [125]
	UKI-1	✓ [125]	n.a.	n.a.	n.a.	n.a.
	Avoralstat	✓ [120]	n.a.	n.a.	n.a.	✓ [120]
	PCI-27483	✓ [120]	n.a.	n.a.	n.a.	✓ [120]
Peptide & Protein	Antipain	✓ [120]	n.a.	n.a.	n.a.	✓ [120]
	Aprotinin	n.a.	✓ [118,143]	n.a.	n.a.	✓ [13,119,149]
	SBTI	✓ [120]	🗴 [143]	n.a.	n.a.	✓ [120]
	α_1_AT	✓ [123,124]	n.a.	n.a.	n.a.	✓/🗴 [123,124,149]
	HAI-2	✓ [135]	✓ [153]	n.a.	n.a.	n.a.
Peptidomimetic	MI-001 (BAPA)	✓ [174]	✓ [57,175]	n.a.	n.a.	n.a.
	MI-432	✓ [174]	✓ [174,175]	n.a.	n.a.	✓ [13]
	MI-1900	✓ [174]	n.a.	n.a.	n.a.	✓ [13]

## 5. Clinical Evaluation of TMPRSS2 Inhibitors in COVID-19 Patients

Very early in the SARS-CoV-2 pandemic, TMPRSS2 was proposed as a potential drug target due to its important role during viral entry, and many clinical studies have been initiated to investigate the efficacy of these drugs in COVID-19 patients. The vast majority of clinical trials focus on inhibitors of TMPRSS2 protease activity (Table 2). However, several retrospective studies have analyzed cohorts of COVID-19 patients under androgen deprivation therapy (ADT). Patients undergoing ADT have been reported to express lower levels of TMPRSS2, and may be less affected by COVID-19 [103]. Montopoli and colleagues reported an association between ADT and reduced occurrence of SARS-CoV-2 infection, whereas others found no beneficial effect of ADT on SARS-CoV-2 infection or COVID-19 progression [177,178,179,180,181,182]. However, the fact that the study cohorts consisted of men suffering from prostate cancer limits the applicability of the results to non-cancer patients with COVID-19. Furthermore, depletion of androgens has broad systemic consequences; therefore, effects cannot be attributed solely to reduction of TMPRSS2 expression.

By the time this review was prepared, the results of five clinical studies and several case reports investigating the therapeutic efficacy of TMPRSS2 inhibitors in COVID-19 patients were available. In a retrospective case series of eleven critically ill COVID-19 cases with organ failure, the clinical improvement of six patients receiving 600 mg (3 × 200 mg) CM per day over the course of five days was compared to a group of five patients receiving hydroxychloroquine [183]. Treatment with orally administered CM was associated with an improvement in the SOFA (Sepsis-related Organ Failure Assessment) score within eight days and a concurrent reduction of inflammatory markers [183]. In line with this, three further case series reported positive impacts of the related TMPRSS2 inhibitor NM on COVID-19 progression [184,185,186]. Unlike CM, NM requires intravenous administration, while the safety and tolerability of oral NM administration is currently under investigation (NCT04406415). Doi and team reported clinical improvement in COVID-19 patients admitted to the intensive care unit (ICU) upon intravenous infusion of NM (0.2 mg/kg/h for 14 days) in combination with administration of the nucleoside analogue Favipiravir. In three COVID-19 cases in South Korea, administration of NM (200 mg for 24 h) followed by CM for four days (600 mg/day) coincided with clinical improvement [185]. Although these case reports are encouraging, they must be interpreted with caution because the small number of patients or lack of a placebo control preclude statistical analysis. Therefore, it remains unclear whether clinical improvement is due to treatment or to natural clearance of infection [183].

Gunst et al. investigated the efficacy of CM treatment in a placebo-controlled phase IIa clinical trial enrolling 208 hospitalized COVID-19 patients [187]. However, administration of 600 mg CM per day over five days had no significant effect on clinical outcomes in comparison to the placebo group [187]. This lack of efficacy could be due to belated administration of the drug or to an insufficient concentration of the drug at the site of viral replication [187]. Kitagawa et al. analyzed the administration of 2400 mg of CM per day in healthy individuals and reported that the drug was well tolerated [188]. Based on this report, a study administering 2400 mg CM in COVID-19 patients was conducted. However, the results are still pending (NCT04657497). A phase II clinical trial of NM was conducted in 104 hospitalized patients who suffered from COVID-19 pneumonia and required oxygen supplementation [189]. Overall, intravenous administration of NM (4.8 mg/kg/day for 10 days) failed to improve the time to clinical improvement and recovery compared with the SOC group [189]. However, a subgroup analysis of patients with worsened baseline clinical status revealed more rapid clinical improvement and a higher recovery rate in the study population treated with NM [189]. Larger and thus more heterogenous study populations will be required to verify the subgroup analysis [189]. Gunst et al. reported a trend towards a lower rate of ICU admission in the CM cohort, which needs confirmation in larger studies [187]. No or only mild adverse events were reported for administration of CM and NM, respectively [187,189].

Several studies have investigated the clinical efficacy of BHH in the context of SARS-CoV-2 infection or COVID-19 progression. In one trial, SOC treatment (*n* = 39) was compared with additional treatment with 24 mg of BHH per day for 14 days (*n* = 39) in patients with COVID-19 symptoms, and a significant decrease in the rate of ICU admission and need for ventilation in the BHH arm was reported [190]. However, in a later study, administration of 32 mg BHH per day for 14 days in 48 hospitalized COVID-19 patients failed to accelerate clinical improvement [191]. Li et al. reported a non-significant trend towards improved clinical recovery in a BHH treatment cohort (*n* = 12) upon administration of the maximum recommended dose of 96 mg per day for 14 days [192].

The inconsistent findings of the aforementioned studies may be due to differences in study design. First, the inclusion criteria varied; enrollment occurred based on COVID-19 symptoms and chest imaging only, in combination with confirmed SARS-CoV-2 infection or a mix thereof. Second, SOC therapy at the study site was different depending on local and/or temporal differences, which could affect the rate of clinical improvement. Third, the definitions and assessment of the primary clinical outcomes were not uniform, variously observing a decline in the clinical score, rate of ICU admission and mortality, or reduction in clinical symptoms [190,191,192]. Overall, larger clinical trials will be required to clarify the impact of BHH on COVID-19 [192].

In a non-peer-reviewed study, Mikhaylov and colleagues analyzed whether prophylactic BHH intake protects against initial SARS-CoV-2 infection [193]. To this end, 25 members of clinical staff having regular contact with suspected COVID-19 patients were compared with 25 staff members treated with daily doses of 24 mg BHH [193]. The group observed no significant changes in the rate of SARS-CoV-2 infection between the two arms but significantly fewer symptomatic cases of COVID-19 in the BHH-treated group [193]. However, the total number of infections (total of *n* = 9) may have been too low to draw definitive conclusions.

In sum, none of the TMPRSS2 inhibitors clearly curbed SARS-CoV-2 infection or mitigated the progression of COVID-19. This could be due to insufficient activity, under-dosing of the drugs, underpowered studies, or belated administration. Importantly, the clinical course of a SARS-CoV-2 infection and the timing of treatment will have a major impact on the success of antivirals. The peak viral load of SARS-CoV-2 is reached before or simultaneously with the onset of symptoms [194,195,196], whereas the main morbidity of COVID-19 is a consequence of the hyperinflammatory and hypercoagulable state, and is associated with less viral replication [197,198]. Thus, the therapeutic window to prevent replication and spread of SARS-CoV-2 is relatively short and direct acting antivirals, such as TMPRSS2 inhibitors, should be administered as early as possible, while treatment with anti-inflammatory compounds can be initiated during later stages of the disease [195,199].

**Table 2 ijms-23-01351-t002:** Overview of initiated and completed clinical studies on TMPRSS2 inhibitors for SARS-CoV-2 infection. CM: Camostat mesylate; NM: Nafamostat mesylate; BHH: Bromhexine hydrochloride; b.i.d.: bies in die (twice a day); t.i.d.: ter in die (three times a day); q.i.d.: quarter in die (four times a day), i.v. intravenously; ICU: intensive care unit; CRP: C-reactive Protein.

Study Title	Identifier, Status	Inclusion Criteria	Intervention	Primary Outcome
A Trial Looking at the Use of Camostat to Reduce Progression of Symptoms of Coronavirus (COVID-19) in People Who Have Tested Positive. (SPIKE-1)	NCT04455815, Active, not recruiting	Confirmed SARS-CoV-2 infection, symptomatic	CM orally, 200 mg q.i.d., 14 d	Safety and efficacy of CM
Camostat Mesylate in COVID-19 Outpatients	NCT04353284, Completed	Confirmed SARS-CoV-2 infection, enrolled within 3 d, symptomatic	CM orally, 200 mg q.i.d., 7 d	Change in SARS-CoV-2 viral load
Camostat Efficacy vs. Placebo for Outpatient Treatment of COVID-19 (CAMELOT)	NCT04583592, Completed	Confirmed SARS-CoV-2 infection, symptomatic, risk for severe illness	CM orally, 200 mg q.i.d., 14 d	Disease progression (hospital or death within 28 d)
Camostat Mesilate Treating Patients With Hospitalized Patients With COVID-19 (RECOVER)	NCT04470544, Recruiting	Confirmed SARS-CoV-2 infection, hospitalized	CM, 200 mg q.i.d.	Proportion of patients alive and free from respiratory failure within 28 d
The Utility of Camostat Mesylate in Patients With COVID-19 Associated Coagulopathy (CAC) and Cardiovascular Complications	NCT04435015, Not yet recruiting	Positive SARS-CoV-2 test, COVID-19 associated coagulopathy/cardiac complication	CM orally, 200 mg, t.i.d.	Percent change in plasma D-Dimer within 7 d
Evaluation of Efficacy and Safety of Camostat Mesylate for the Treatment of SARS-CoV-2 Infection—COVID-19 in Ambulatory Adult Patients (CAMOVID)	NCT04608266, Recruiting	PCR confirmed SARS-CoV-2 infection, symptomatic, risk of severe COVID-19	CM orally, 200 mg, t.i.d., 14 d	Hospitalization for COVID-19 deterioration or death
Oral Camostat Compared With Standard Supportive Care in Mild-Moderate COVID-19 Patients (COPS-2003)	NCT04524663, Completed	Confirmed SARS-CoV-2 infection, symptomatic, mild to moderate COVID-19	CM orally, 10 d	Time until cessation of shedding of SARS-CoV-2
The Potential of Oral Camostat in Early COVID-19 Disease in an Ambulatory Setting to Reduce Viral Load and Disease Burden	NCT04625114, Recruiting	PCR confirmed SARS-CoV-2 infection, signs or symptoms of COVID-19, outpatient	CM orally, 100 mg t.i.d., 5–10 d	Reduction of viral load (RT qpCR)
The DAWN Camostat Trial for Ambulatory COVID-19 Patients	NCT04730206, Recruiting	Positive SARS-CoV-2 antigen test, symptomatic < 5 d, outpatient	CM orally, 200 mg q.i.d., 7 d	Time to recovery, unplanned hospitalization or death
The Impact of Camostat Mesilate on COVID-19 Infection (CamoCO-19)	NCT04321096, [187]	PCR confirmed SARS-CoV-2 infection, hospitalized, symptomatic < 48 h	CM orally, 200 mg t.i.d., 5 d	Time to clinical improvement
Multiple-dose Study of FOY-305 in Japanese Healthy Adult Male Subjects	NCT04451083, Completed	Healthy adults	CM orally, multiple doses q.i.d.	Safety and tolerability
COVID-19 Outpatient Pragmatic Platform Study (COPPS)-Camostat Sub-Protocol (COPPS)	NCT04662073, Active not recruiting	Confirmed SARS-CoV-2 infection, enrolled within 72 h, mild-moderate COVID-19	CM orally, 200 mg q.i.d.	Change in SARS-CoV-2 shedding
Reconvalescent Plasma/Camostat Mesylate Early in SARS-CoV-2 Q-PCR (COVID-19) Positive High-risk Individuals (RES-Q-HR)	NCT04681430, Recruiting	PCR confirmed SARS-CoV-2 infection, enrolled within 3 d, symptomatic COVID-19	CM orally, 200 mg t.i.d., 7 d	Clinical status improvement
A Study of FOY-305 in Patients With SARS-CoV-2 Infection (COVID-19)	NCT04657497, Completed	Confirmed SARS-CoV-2 infection, symptomatic, enrolled within 5 d after symptom onset	CM orally, 600 mg q.i.d., 14 d	Positivity of SARS-CoV-2 PCR
Safety and Pharmacokinetics Evaluation Study According to the Dose of Camostat Mesylate in Healthy Volunteers	NCT04782505, Recruiting	Healthy adults	CM orally, 100–300 mg	Pharmacokinetics
COVID-19 Outpatient Pragmatic Platform Study (COPPS)—Master Protocol	NCT04662086, Recruiting	Confirmed SARS-CoV-2 infection, enrolled within <72 h, mild-moderate COVID-19	CM orally, 200 mg q.i.d.	Change in SARS-CoV-2 shedding Time to sustained symptom resolution
Novel Agents for Treatment of High-risk COVID-19 Positive Patients	NCT04374019, Recruiting	Confirmed SARS-CoV-2 infection, high risk feature for clinical deterioration	CM orally, 200 mg t.i.d., 14 d	Clinical deterioration
ACTIV-2: A Study for Outpatients With COVID-19	NCT04518410, Recruiting	Confirmed SARS-CoV-2 infection symptomatic, enrolled within 7 d	CM orally, 200 mg t.i.d., 7 d	COVID-19 symptom duration and SARS-CoV-2 viral load
Efficacy and Safety of DWJ1248 in Patients With Mild to Moderate COVID-19 Compared to the Placebo	NCT04521296, Active, not recruiting	Confirmed SARS-CoV-2 infection, mild or moderate COVID-19	CM orally, 200 mg t.i.d.	SARS-CoV-2 viral load, clinical improvement
A Study of DWJ1248 in Prevention of COVID-19 Infection After the Exposure of SARS-CoV-2	NCT04721535, Not yet recruiting	Contact with SARS-CoV-2 positive patient, asymptomatic and PCR negative subjects	CM orally, 200 mg 1x daily	SARS-CoV-2 positivity (RT-PCR)
Efficacy and Safety of DWJ1248 With Remdesivir in Severe COVID-19 Patients	NCT04713176, Recruiting	PCR confirmed SARS-CoV-2 infection, enrolled within 10 d	CM orally, 200 mg 1x daily, up to 14 d Remdesivir	Mortality rate or ECMO patients
Clinical Efficacy of Nafamostat Mesylate for COVID-19 Pneumonia	NCT04418128, Not yet recruiting	Confirmed SARS-CoV-2 infection, hospitalized, pneumonia within 72 h, no oxygenation	NM i.v. 0.1–0.2 mg/kg/h over 24 h, up to 14 d	Clinical improvement
Efficacy of Nafamostat in COVID-19 Patients (RACONA Study) (RACONA)	NCT04352400, Recruiting	Confirmed SARS-CoV-2 infection, hospitalized	NM i.v.	Time to clinical improvement
Oral Nafamostat in Healthy Volunteers	NCT04406415, Completed	Healthy adults	NM orally, 10–200 mg t.i.d., 5 d	Safety and tolerability
Efficacy and Safety Evaluation of Treatment Regimens in Adult COVID-19 Patients in Senegal (SEN-CoV-Fadj)	NCT04390594, Recruiting	Confirmed SARS-CoV-2 infection, pneumonia within 72 h of symptom onset	NM i.v. 0.1–0.2 mg/kg/h, up to 14 d	SARS-CoV-2 viral load
Australasian COVID-19 Trial (ASCOT) ADAptive Platform Trial (ASCOT ADAPT)	NCT04483960, Recruiting	Confirmed SARS-CoV-2 infection, symptomatic, hospitalized	NM i.v. 0.2 mg/kg/h, 7 d	Mortality rate or requirement of ICU support
DEFINE—Evaluating Therapies for COVID-19 (DEFINE)	NCT04473053, Active, not recruiting	Confirmed SARS-CoV-2 infection, symptomatic	NM 0.2 mg/kg/h, 7 d	Safety and tolerability
A Study Evaluating the Efficacy and Safety of CKD-314 (Nafabelltan) in Hospitalized Adult Patients Diagnosed With COVID-19 Pneumonia	NCT04623021, [189]	Confirmed SARS-CoV-2 infection, pneumonia	NM i.v. 4.8 mg/kg/day, 10 d	Time to clinical improvement
A Study Evaluating the Efficacy and Safety of CKD-314 in Hospitalized Adult Patients Diagnosed With COVID-19 Pneumonia	NCT04628143, Completed	Confirmed SARS-CoV-2 infection, pneumonia, hospitalized	NM i.v.	Time to clinical improvement
Phase 3 Clinical Trial to Evaluate the Efficacy and Safety of CKD-314	NCT04871646, Recruiting	Confirmed SARS-CoV-2 infection, pneumonia	NM i.v.	Time to recovery
Use of Bromhexine and Hydroxychloroquine for Treatment of COVID-19 Pneumonia	NCT04355026, Recruiting	PCR confirmed SARS-CoV-2 infection, hospitalized	BHH orally, 16 mg t.i.d.	Duration of hospitalization and disease
BromhexIne And Spironolactone For CoronaVirUs Infection Requiring HospiTalization (BISCUIT)	NCT04424134, Recruiting	Confirmed SARS-CoV-2 infection, pneumonia	BHH, 8 mg q.i.d., 10 d Spironolactone 50 mg, 10 d	Change from baseline in clinical score
Clinical Trial With N-acetylcysteine and Bromhexine for COVID-19	NCT04928495, not yet recruiting	Clinical signs and symptoms of COVID-19	BHH 32 mg/day 10 d N-acetylcysteine 1800 mg/day, 10 d	Time of recovery
Evaluating the Efficacy and Safety of Bromhexine Hydrochloride Tablets Combined with Standard Treatment/Standard Treatment in Patients With Suspected and Mild Novel Coronavirus Pneumonia (COVID-19)	NCT04273763, [97]	Confirmed or suspected mild or moderate COVID-19	BHH orally, 32 mg t.i.d., 14 d	Time of clinical recovery and deterioration rate
Prevention of Infection and Incidence of COVID-19 in Medical Personnel Assisting Patients with New Coronavirus Disease	NCT04405999, [193], preprint	Medical personnel at risk for COVID-19, negative PCR for SARS-CoV-2 infection	BHH	SARS-CoV-2 viral load
Study on the Pharmacokinetics of Bromine Hexane Hydrochloride Tablets in Healthy Adults	NCT04672707, not yet recruiting	Healthy adults	BHH orally, 32–80 mg t.i.d., 2 d	Pharmacokinetics
Effect of bromhexine on clinical outcomes and mortality in COVID-19 patients: A randomized clinical trial	IRCT202003117046797N4, [190]	Diagnosed COVID-19 pneumonia	BHH orally, 8 mg t.i.d., 14 d	Improvement in rate of ICU admission, intubation and ventilation, 28-day mortality
Effect of bromhexine in hospitalized patients with COVID-19	[191]	PCR confirmed SARS-CoV-2 infection, hospitalized	BHH orally, 8 mg q.i.d., 14 d	Clinical improvement
An Open Non-comparative Study of the Efficacy and Safety of Aprotinin in Patients Hospitalized with COVID-19	NCT04527133, active, not recruiting	PCR confirmed SARS-CoV-2 infection, moderate to severe COVID-19	Stage 1: Aprotinin i.v., 1,000,000 KIU IV, 3 d Stage 2: Aprotinin inhaled, 625 KIU q.i.d., 5 d or: Aprotinin i.v. 1,000,000 KIU IV 1x daily, 5 d + Favirapir	Time to SARS-CoV-2 negativity, CRP and D-Dimer normalization
Study to Evaluate the Safety and Efficacy of Prolastin in Hospitalized Subjects with COVID-19	NCT04495101, completed	PCR confirmed SARS-CoV-2 infection, symptomatic, hospitalized	α_1_AT i.v., 120 mg/kg, day 1 and day 8	Percentage of subjects dying and dependent on ventilation
Trial of Alpha One Antitrypsin Inhalation in Treating Patient with Severe Acute Respiratory Syndrome Coronavirus 2 (SARS-CoV-2)	NCT04385836, Recruiting	PCR confirmed SARS-CoV-2 infection, hospitalized	α_1_AT inhalative, b.i.d., 5 d	Clinical improvement
Study to Evaluate the Safety and Efficacy of Liquid Alpha1-Proteinase Inhibitor (Human) in Hospitalized Participants with Coronavirus Disease (COVID-19)	NCT04547140, recruiting	PCR confirmed SARS-CoV-2 infection, symptomatic, hospitalized	α_1_AT i.v., 120 mg/kg, day 1 and day 8	Percentage of subjects dying and dependent on ventilation
A randomized double-blind placebo-controlled pilot trial of intravenous plasma-purified alpha-1 antitrypsin for severe COVID-19 illness.	EudraCT 2020-001391-15, Completed	Confirmed COVID-19 infection, moderate ARDS	α_1_AT, i.v. 120 mg/kg weekly, 28 d	Biological activity of A1AT as anti-inflammatory therapy

## 6. Conclusions

TMPRRS2 plays a major role in the entry of several respiratory viruses and represents a promising therapeutic target. Various small molecule, peptide and protein inhibitors of TMPRSS2 have been identified which efficiently block influenza and coronavirus infection and replication in vitro and in animal models. At the moment, clinical studies are primarily investigating the anti-SARS-CoV-2 activity of TMPRSS2 inhibitors, with a focus on symptomatic and/or hospitalized COVID-19 patients. In clinical studies to date, failure to observe reproducible therapeutic effects may be due to the late administration of TMPRSS2 inhibitors. Thus, studies examining the effects of early treatment with TMPRSS2 inhibitors are highly warranted. Consequently, targeted administration of TMPRSS2 inhibitors via inhalation might suppress SARS-CoV-2 more efficiently than systemic administration. Targeting multiple steps of the viral replication cycle by combining TMPRSS2 inhibitors with nucleoside analoga or neutralizing antibodies poses an additional promising strategy in the search for effective anti-SARS-CoV-2 therapies. In conclusion, the potential of TMPRSS2-inhibiting antivirals against respiratory viruses clearly warrants further research.

## Figures and Tables

**Figure 1 ijms-23-01351-f001:**
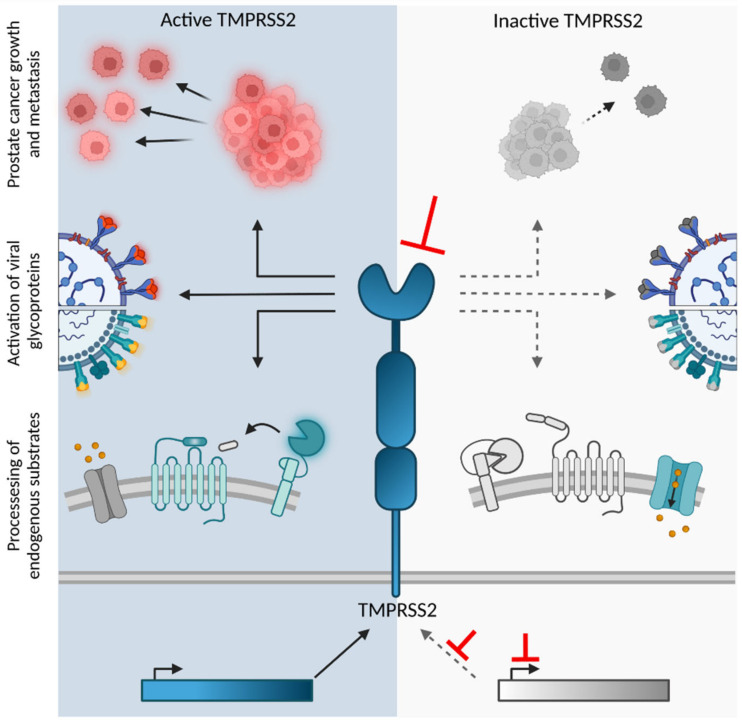
(Patho)-physiological role of TMPRSS2. Left panel: the protease activity of TMPRSS2 contributes to the activation of viral glycoproteins, the progression of prostate cancer, and the cleavage of endogenous substrates such as the epithelial sodium channel, TTSP zymogens, and protease activated receptors. Inhibition of TMPRSS2 expression or protease activity (red arrows, right panel) results in reduced prostate cancer growth and viral entry, although it may also lead to reduced cleavage of endogenous substrates. Figure created with BioRender.com (accessed on 13 December 2021).

**Figure 2 ijms-23-01351-f002:**
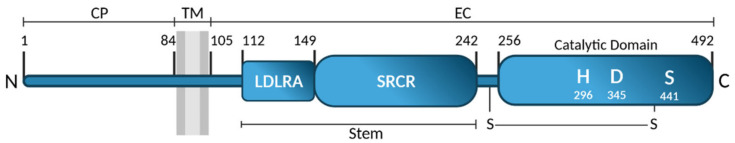
Architecture of TMPRSS2. CP: cytoplasmatic domain; TM: transmembrane domain; EC: extracellular domain; LDLRA: low density lipoprotein receptor class A domain; SRCR: scavenger receptor cysteine-rich domain; H, D, S: catalytic triad (histidine, aspartate, serine); N and C: N- and C-terminus, respectively. S-S indicates disulfide bridge (note that only the disulfide bridge between the catalytic and SRCR domains is shown, with the remaining disulfide bridges omitted for clarity). Numbers indicate amino acid position according to UniProt O15393. Figure created with BioRender.com (accessed on 13 December 2021).

**Figure 3 ijms-23-01351-f003:**
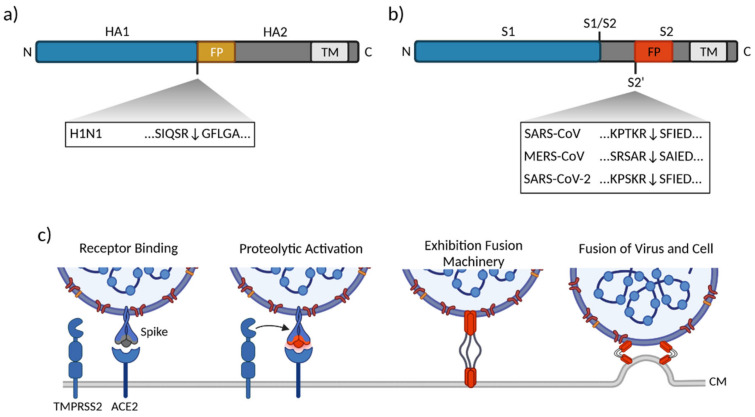
Schematic representation of (**a**) influenza virus hemagglutinin (HA) and (**b**) coronavirus spike (S) glycoproteins. FP: fusion peptide, TM: transmembrane domain. Downward arrows indicate TMPRSS2 cleavage sites. Sequences derived from Uniprot Q9WFX3 (H1N1); P59594 (SARS-CoV); A0A023SFE5 (MERS-CoV); P0DTC2 (SARS-CoV-2). (**c**) Schematic representation of SARS-CoV-2 entry mediated by TMPRSS2. Figure created with BioRender (accessed on 13 December 2021).

**Figure 4 ijms-23-01351-f004:**
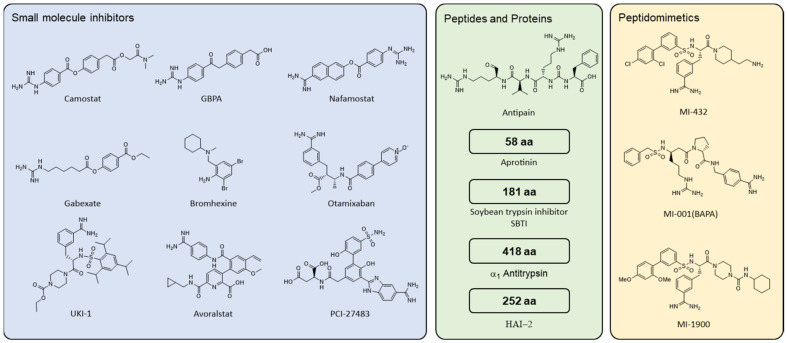
Overview of inhibitors of TMPRSS2 protease activity.

## Data Availability

Not applicable.

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
