# Peer review of "The Transmembrane Protease TMPRSS2 as a Therapeutic Target for COVID-19 Treatment"

_ijms, 2022, doi:10.3390/ijms23031351_

Round 1

Reviewer 1 Report

An excellent review. Needs minor idiomatic editing.

Author Response

We thank the reviewer for appreciating our efforts and tried to improve the text throughout the manuscript. The text has been edited by a native English speaker throughout.

Reviewer 2 Report

Wettstein and co-authors review the role of TMPRSS2, a protease implicated in very different actions, such as prostate cancer progression or viral entry.

However, after a careful reading, one may surmise that authors' scope is only to describe the use of TMPRSS2 inhibitors in COVID infection. The review may lead to some misunderstanding in the first four pages and makes the manuscript sounds lopsided. The two main reasons are as follows:

  1. the title does not reflect the main scope of the manuscript. The title should be clearer when specifying the interest of the authors in blocking TMPRSS2 actions to block viral entry and mainly SARS-CoV2.
  2. mentioning the role of TMPRSS2 in the progression of prostate cancer may mislead the reader as the manuscript will continue with viral entry and the inhibitors (a quarter of the manuscript are only inhibitors descriptions) and no further information is offered regarding cancer.

The manuscript flows unclearly in current form. For this reason, the reviewer recommends rewriting the first four pages of the manuscript as introductory text and then describing in deep TMPRSS2 inhibitors to block viral entry.

Minor comments:

Authors should expand the abstract for clarity for the readers. 

Graphical abstract and Figure 1 are redundant. Probably, Figure 1 will be a better graphical abstract after highlighting the viral entry in some way and figure 1 should be removed from the introduction.

Line 69 should be linked with line 85 regarding hormone control.

Author Response

Reviewer 2

We thank reviewer 2 for the evaluation of our paper and the helpful comments.

Wettstein and co-authors review the role of TMPRSS2, a protease implicated in very different actions, such as prostate cancer progression or viral entry.

However, after a careful reading, one may surmise that authors' scope is only to describe the use of TMPRSS2 inhibitors in COVID infection. The review may lead to some misunderstanding in the first four pages and makes the manuscript sounds lopsided.

The two main reasons are as follows:

The title does not reflect the main scope of the manuscript. The title should be clearer when specifying the interest of the authors in blocking TMPRSS2 actions to block viral entry and mainly SARS-CoV2.

We have changed the title to “The transmembrane protease TMPRSS2 as a therapeutic target for COVID-19 treatment”

Mentioning the role of TMPRSS2 in the progression of prostate cancer may mislead the reader as the manuscript will continue with viral entry and the inhibitors (a quarter of the manuscript are only inhibitors descriptions) and no further information is offered regarding cancer.

Also in accordance with reviewer 3, we omitted the detailed description of the role of TMPRSS2 in prostate cancer and now only focus on the antiviral aspects of TMPRSS2 inhibition.

The manuscript flows unclearly in current form. For this reason, the reviewer recommends rewriting the first four pages of the manuscript as introductory text and then describing in deep TMPRSS2 inhibitors to block viral entry.

Also in accordance with reviewer 3, we restructured the first chapters, deleted the paragraphs on prostate cancer. Moreover the text was edited by a native English speaker.

Minor comments:

Authors should expand the abstract for clarity for the readers.

We expanded the abstract accordingly.

Graphical abstract and Figure 1 are redundant. Probably, Figure 1 will be a better graphical abstract after highlighting the viral entry in some way and figure 1 should be removed from the introduction.

To our understanding, the graphical abstract is not part of the downloaded PDF version of the review, therefore we would like to keep the more detailed figure 1 in the main text and the graphical abstract as online material.

Line 69 should be linked with line 85 regarding hormone control.

Not applicable due to changes in the text.

Reviewer 3 Report

Lukas Wettstein and colleagues presented a comprehensive review article aimed at describing the pathophysiological role of transmembrane serine protease (TMPRSS2) with particular reference to viral infection and COVID-19 infection. Overall, the manuscript is well-structured and well-conceived, however, there are some parts that should be removed and some details that the authors have to add. Below are reported some minor comments that will improve the quality of the manuscript:
1) The authors have to add the clinical context in the title (add COVID-19, SARS-CoV-2 or something similar); 
2) Check the grammar of the following sentence: “Addition of the protease induced activation of PAR-2 (but not PAR-1) and resulted in calcium ion release from intracellular stores, indicative for initiation of downstream signaling [30,31].”; 
3) Throughout the manuscript there are some errors. Please check the entire text and perform English editing;
4) As the main topic of the review is the role of TMPRSS2 in viral infections, I suggest to remove the chapter “3.1. Prostate Cancer”. Consider to remove the description of prostate cancer also in the other chapters (e.g. Chapter 4);
5) Please provide more supporting references for the following statements: “Of note, next to TMPRSS2, several TTSPs were able to prime SARS-CoV-2 S in vitro. Based on their expression levels, TMPRSS11D, TMPRSS11E and TMPRSS11F as well as TMPRSS13 are unlikely to act in the lung, but might contribute to extrapulmonary SARS-CoV-2 spread [88]. It would be intriguing to gain more insight into the importance of these proteases on SARS-CoV-2 infection.”. For this purpose, please see:
- PMID: 34935057
- PMID: 32979398
- PMID: 33649798

Author Response

Reviewer 3

We thank reviewer 3 for his/her evaluation of our manuscript and the helpful comments.

Lukas Wettstein and colleagues presented a comprehensive review article aimed at describing the pathophysiological role of transmembrane serine protease (TMPRSS2) with particular reference to viral infection and COVID-19 infection. Overall, the manuscript is well-structured and well-conceived, however, there are some parts that should be removed and some details that the authors have to add.

Below are reported some minor comments that will improve the quality of the manuscript:

The authors have to add the clinical context in the title (add COVID-19, SARS-CoV-2 or something similar);

We have changed the title to “The transmembrane protease TMPRSS2 as a therapeutic target for COVID-19 treatment”

  • Check the grammar of the following sentence: “Addition of the protease induced activation of PAR-2 (but not PAR-1) and resulted in calcium ion release from intracellular stores, indicative for initiation of downstream signaling [30,31].”;

Corrected

  • Throughout the manuscript there are some errors. Please check the entire text and perform English editing;

Done.

  • As the main topic of the review is the role of TMPRSS2 in viral infections, I suggest to remove the chapter “3.1. Prostate Cancer”. Consider to remove the description of prostate cancer also in the other chapters (e.g. Chapter 4);

We omitted the chapter on prostate cancer (3.1).

  • Please provide more supporting references for the following statements: “Of note, next to TMPRSS2, several TTSPs were able to prime SARS-CoV-2 S in vitro. Based on their expression levels, TMPRSS11D, TMPRSS11E and TMPRSS11F as well as TMPRSS13 are unlikely to act in the lung, but might contribute to extrapulmonary SARS-CoV-2 spread [88]. It would be intriguing to gain more insight into the importance of these proteases on SARS-CoV-2 infection.”. For this purpose, please see:

- PMID: 34935057

- PMID: 32979398

- PMID: 33649798

We added the respective papers.

Round 2

Reviewer 2 Report

Authors addressed reviewer's concerns accordingly.